# Cutaneous Sarcoidosis-like Eruption Following Second Dose of Moderna mRNA-1273 Vaccine: Case or Relationship?

**DOI:** 10.3390/diagnostics13071286

**Published:** 2023-03-29

**Authors:** Gerardo Cazzato, Francesca Ambrogio, Caterina Foti, Marialessandra Capuzzolo, Irma Trilli, Nadia Casatta, Carmelo Lupo, Marianna Carrieri, Daniele Daini, Anna Colagrande, Eugenio Maiorano, Giuseppe Ingravallo

**Affiliations:** 1Section of Molecular Pathology, Department of Precision and Regenerative Medicine and Ionian Area (DiMePRe-J), University of Bari “Aldo Moro”, 70124 Bari, Italy; 2Section of Dermatology and Venereology, Department of Precision and Regenerative Medicine and Ionian Area (DiMePRe-J), University of Bari “Aldo Moro”, 70124 Bari, Italy; 3Odontomatostologic Clinic, Department of Innovative Technologies in Medicine and Dentistry, University of Chieti “G. D’Annunzio”, 66100 Chieti, Italy; 4Innovation Department, Diapath S.p.A., Via Savoldini n.71, 24057 Martinengo, Italy; 5Independent Researcher, 70124 Bari, Italy; 6Dermatology and Venereology, ASL Lecce, 73100 Lecce, Italy

**Keywords:** SARS-CoV-2, vaccines, adverse drug reactions, cutaneous, sarcoidosis, sarcoid-like

## Abstract

Various adverse reactions to SARS-CoV-2 vaccines have been described since the first months of the vaccination campaign. In addition to more frequent reactions, rare reactions, such as sarcoidosis-like, rashes have been reported. We present a case of a 23-year-old woman with a rash on the chin and peribuccal region, which developed approximately 3 weeks after the administration of the second dose of the Moderna mRNA-1273 vaccine. We briefly discuss other reports in the literature.

**Figure 1 diagnostics-13-01286-f001:**
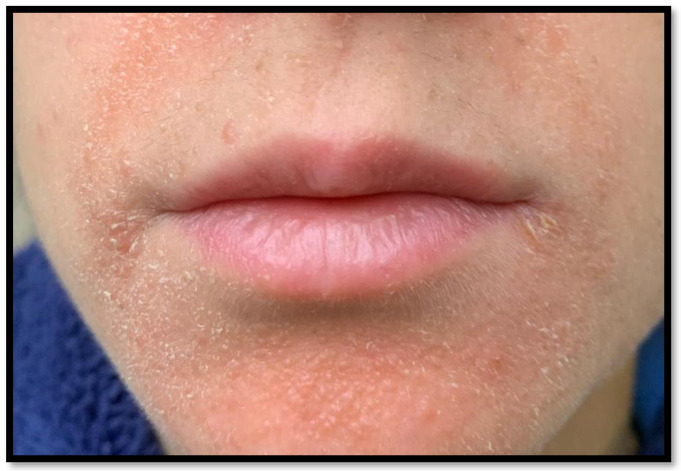
Clinical picture showing yellow/orange papules present in the chin and peribuccal region of a 23-year-old woman 3 weeks after being vaccinated with the second dose of anti-SARS-CoV-2 vaccine. A 23-year-old girl presented to her dermatologist, on the recommendation of her general practitioner, for the appearance of a papular, yellowish rash on the nose, chin and peribuccal area, which appeared approximately 3 weeks after the administration of the second dose of Moderna mRNA-1273 vaccine. First clinically diagnosed with, and then treated for, seborrhoeic dermatitis, the patient did not report any improvement; thus, treatment with saline compresses was started with slight improvement for a few days. Following the persistence of this clinical picture, the dermatologist used off label Zelaic acid and Metronidazole without any improvement. Following this, a serum determination of angiotensin-converting enzyme 2 (ACE2) was requested, which was found to be within the normal range (89 U/L). In the meantime, diascopy was performed on some of the lesions, from which characteristic yellow/orange colouring emerged. The approval and large-scale dissemination of anti-SARS-CoV-2 vaccines has allowed restrictions to be relaxed worldwide and allowed a return to more normal lifestyles similar to the pre-pandemic era [1]. Since the first months of the vaccination campaign, an increasing number of adverse reactions to COVID-19 vaccines have been reported, both in the case of mRNA vaccines (BioNTech-Pfizer, BNT162b2 and Moderna, mRNA-1273) and adenovirus vector vaccines (Vaxzevria and Jcovden) [2,3]. According to a recent study by Gambicher et al. (who reviewed the literature and the results of the most clinical trials), adverse skin reactions from the anti-SARS-CoV-2 vaccine were subdivided, according to the time of onset, into early and late onset. Early local reactions at the vaccination site (type I) included pain (88%), pruritus (35%), induration (25%), erythema (20%) and oedema (15%), and late-onset reactions (type IV), which occur after about 1 week, included erythema, induration, pain, inflammatory reactions in a dermal filler or past the radiation areas, old BCG scars, and, more frequently, morbilliform and erythema multiforme-like rashes [4]. Rare conditions were also more present, such as autoimmune-mediated skin diseases, including leucocytoclastic vasculitis, lupus erythematosus and immune thrombocytopenia, and functional angiopathies, such as chilblain-like lesions and erythromelalgia [4]. Finally, there have also been reports of herpes zoster recurrence and pityriasis rosea-like rashes following COVID-19 vaccination [5,6]. In this regard, there have been rare and anecdotal reports of sarcoid-like reactions or the unmasking of a previous sarcoidosis in patients receiving anti-SARS-CoV-2 vaccination. In this paper, we present a case of a sarcoid-like reaction on the face of a 23-year-old patient, which occurred 3 weeks after the administration of the second dose of the Moderna mRNA-1273 vaccine, and describe very briefly the other cases presented in the literature.

**Figure 2 diagnostics-13-01286-f002:**
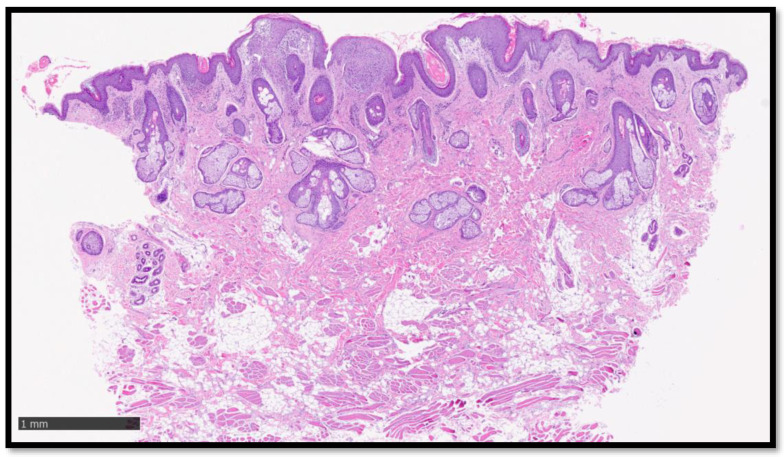
Histological photomicrograph showing the biopsy sample taken and analysed. Note, in the centre and left of the image, the presence of a papillomatous epidermis, beneath which is an inflammatory infiltrate (Hematoxylin-Eosin, original magnification 2×, Diapath^TM^, Martinengo, Italy). It was, therefore, decided that a skin biopsy would be performed at the level of one of these papular lesions in the peribuccal region. Histological analysis showed non-caseating “naked” granuloma, mainly present in the superficial dermis, a level of the dermo–epidermal junction (Figure 1, Figure 2 and Figure 3). These granulomas were characterized by a relatively high number of clusters of epithelioid histiocytes with a poorly developed or totally absent lymphocytic cuff (Figure 4). There were no evident asteroid bodies (also called Schaumann body) (Figure 5).

**Figure 3 diagnostics-13-01286-f003:**
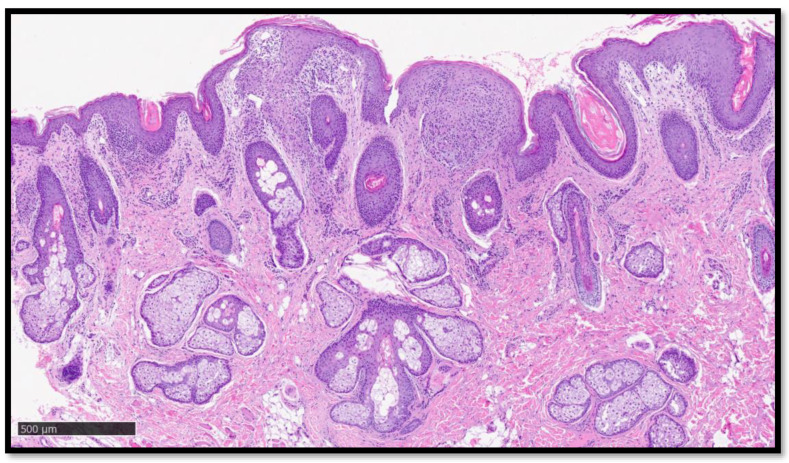
Histological photomicrograph showing, at higher magnification, presence of granulomatous inflammatory infiltrate, mainly localized below the dermo–epidermal junction, partially raised, with the presence of moderately ectasic superficial dermal capillary plexus blood vessels (Hematoxylin-Eosin, original magnification 4×).

**Figure 4 diagnostics-13-01286-f004:**
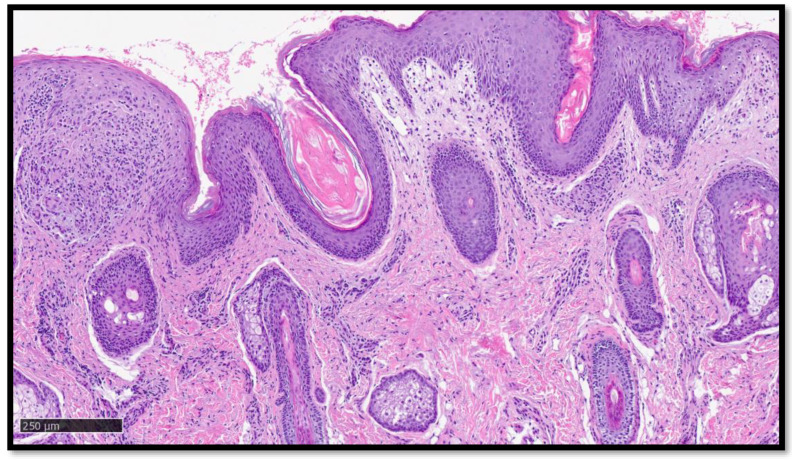
Histological preparation showing a field from Figure 3, in which the gigantocellular granulomatous infiltrate can be seen on the left (Hematoxylin-Eosin, original magnification 10×).

**Figure 5 diagnostics-13-01286-f005:**
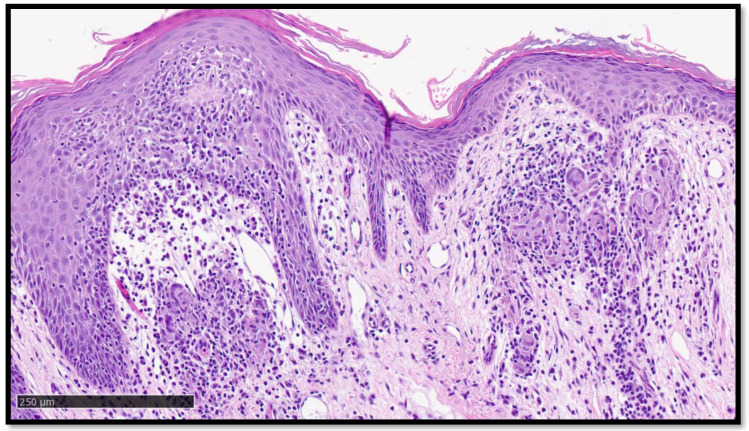
Histological preparation showing multinucleated giant cells with presence of many lymphocytes above the dermo–epidermal junction, with some plasma cells and ectatic vessels (Hematoxylin-Eosin, original magnification 20×). After the histological examination, the dermatologist sent the patient to the pulmonologists in an attempt to rule out a possible misdiagnosed sarcoidosis. A total body CT/PET scan was performed, which showed no suspicious areas or hyperaccumulation. At a 3-month follow-up, the rash had completely resolved. No recurrences have been observed and the patient is in good health. Several adverse effects from the anti-SARS-CoV-2 vaccine have been reported since the beginning of the vaccination campaigns. Currently, reports of sarcoidosis or sarcoid-like reactions following COVID-19 vaccination are rare and anecdotal. The first researchers to describe the occurrence of Lofgren’s syndrome in two vaccinated patients were Rademacher et al. [7], who, in their article, reported the case of a patient vaccinated with ChadOx-1, Astra Zeneca in the first dose and CX-024414, Moderna in the second dose, and who had developed periarthritis of the hip with oedema and consensual tenosynovitis. The patient, initially medicated with ibuprofen at a dose of 600 mg up to 3 times daily, showed no other signs of possible localisation of granulomas, and, after therapy with glucocorticoids, had gone into remission of symptoms. The second patient, described by the authors as a 27-year-old Caucasian man, presented with symptoms 28 days after his first vaccination with ChadOx-1, Astra Zeneca. In this case, there was also enlargement of the hips, bilaterally, with pain and an extensive rash in the legs, and morphological features similar to Erythema Nodosum (EN). In addition, bilateral hilar, mediastinal lymphadenopathy and fine nodular parenchymal changes were present, all of which are consistent with a diagnosis of sarcoidosis. This second patient also resolved their clinical picture rather rapidly. A case of a sarcoid-like manifestation following SARS-CoV-2 pneumonia in a 72-year-old woman has also been described in the literature [8], and a case conducted by Grieco T. et al. reported a similar reaction in a 73-year-old patient [9]. From the few existing articles in the literature, it seems that anti-SARS-CoV-2 vaccines could act as a trigger for sarcoidosis-like reactions, considering that in our case, as in the case of Grieco T. et al., there was no previous sarcoidosis, nor a development of sarcoidosis-like disease, following the administration of the vaccine; rather, there was only skin involvement by a sarcoid inflammatory infiltrate. Possible explanations for this skin manifestation include the likely activation of dendritic and macrophage cells stimulated by pathogen-associated molecular patterns, such as viral RNA-triggered pattern-recognition receptor signalling. In this view, these last types of immune cells could release proinflammatory mediators, such as interleukin 12, allowing CD4+ T-lymphocytes to be polarised towards the T-helper 1 (Th1) phenotype. It is important to emphasise that we have not made a definite link between vaccination and the development of these manifestations, but that we have described an association between these two events [10]. Further studies that, in addition to analyses of the most common adverse skin manifestations, also investigate such rare eruptions are necessary in order to better understand this possible manifestation during the COVID-19 vaccination campaign.

## Data Availability

Not applicable.

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
