# Peer review of "Cutaneous Sarcoidosis-like Eruption Following Second Dose of Moderna mRNA-1273 Vaccine: Case or Relationship?"

_diagnostics, 2023, doi:10.3390/diagnostics13071286_

Round 1

Reviewer 1 Report

The case study submitted by Czatto et al represents a single patient finding sharing adverse side effects from the Covid-19 vaccine. This finding could be anecdotal as there are no other studies that reported such finding outside of Europe. The histology presented in the study requires proper labeling showing multinucleated giant histiocytes and non-caseating granulomatous structures. Additionally, this study could be made more interesting by performing single-cell sequencing on the skin biopsy to identify changes in macrophage and CD4 T cells, and or identify a causative agent. Further, HLA-genotyping results would help better understand genetic association for the development of such kind of response to Covid-19 vaccination.

Author Response

Reviewer n’1: The case study submitted by Czatto et al represents a single patient finding sharing adverse side effects from the Covid-19 vaccine. This finding could be anecdotal as there are no other studies that reported such finding outside of Europe. The histology presented in the study requires proper labeling showing multinucleated giant histiocytes and non-caseating granulomatous structures.

Answer n’1: Dear Reviewer n’1, thank you very much. Really, there are some other reports in literature about this cutaneous manifestation and we highlighted them in our paper. So, regarding the labeling, samples was too small (punch biopsy 3 mm) and there is not enough to perform immunohistochemical investigation. But, like in other papers already published, histological picture (Hematoxylin-Eosin) should be sufficient. Thank you.

Reviewer n’1: Additionally, this study could be made more interesting by performing single-cell sequencing on the skin biopsy to identify changes in macrophage and CD4 T cells, and or identify a causative agent.

Answer n’2: Thank you, but in our institution is not possible to perform a single-cell sequencing. Thanks again.

Reviewer n’1: Further, HLA-genotyping results would help better understand genetic association for the development of such kind of response to Covid-19 vaccination.

Answer n’3: Dear Reviewer n’1, thank you very much. We checked this information and we find that there is not HLA mutation/polymorphism. Thank for all.

Reviewer 2 Report

In the present paper, the authors reported a cutaneous Sarcoidosis-like eruption following second dose of 2 Moderna mRNA-1273 vaccine, suggesting a likely causal relationship of vaccination. Although this is not the first and the only case in literature reporting sarcoid-like reaction after vaccination, it is important to describe this further manifestation. Minor points:

1) The authors mentioned that the dermatologist sent the patient to the pulmonologists in an attempt to rule out a possible misdiagnosed sarcoidosis, but they did not describe in detail the patient diagnostic work-up 

2) the authors should state that it is not possible to establish a certain causal relationship 

3)A similar paper has been recently published ("A Novel Development of Sarcoidosis Following COVID-19 vaccination and literature review" inter med 2022 Oct 15;61(20):3101-3106). This should be cited in the discussion and in the bibliography

Author Response

Reviewer n’2: In the present paper, the authors reported a cutaneous Sarcoidosis-like eruption following second dose of 2 Moderna mRNA-1273 vaccine, suggesting a likely causal relationship of vaccination. Although this is not the first and the only case in literature reporting sarcoid-like reaction after vaccination, it is important to describe this further manifestation.

Answer n’1: Dear Reviewer n’2, thank you very much.

Reviewer n’2: 1) The authors mentioned that the dermatologist sent the patient to the pulmonologists in an attempt to rule out a possible misdiagnosed sarcoidosis, but they did not describe in detail the patient diagnostic work-up.

Answer n’2: Thank you very much for this useful advice. We added these informations. Thanks a lot.

Reviewer n’2: 2) the authors should state that it is not possible to establish a certain causal relationship

Answer n’3: Dear Reviewer n’2, thank you. We stated it.

Reviewer n’2: 3)A similar paper has been recently published ("A Novel Development of Sarcoidosis Following COVID-19 vaccination and literature review" inter med . 2022 Oct 15;61(20):3101-3106). This should be cited in the discussion and in the bibliography.

Answer n’4: Done. Thank you.